# An Accurate Classification of Rice Diseases Based on ICAI-V4

**DOI:** 10.3390/plants12112225

**Published:** 2023-06-05

**Authors:** Nanxin Zeng, Gufeng Gong, Guoxiong Zhou, Can Hu

**Affiliations:** 1College of Computer & Information Engineering, Central South University of Forestry and Technology, Changsha 410018, China; m15271698243@163.com; 2Hunan Polytechnic of Environment and Biology, Hengyang 421005, China; hc@hnebp.edu.cn

**Keywords:** rice disease detection, coordinate attention, ICAI-V4, Candy algorithm, involution

## Abstract

Rice is a crucial food crop, but it is frequently affected by diseases during its growth process. Some of the most common diseases include rice blast, flax leaf spot, and bacterial blight. These diseases are widespread, highly infectious, and cause significant damage, posing a major challenge to agricultural development. The main problems in rice disease classification are as follows: (1) The images of rice diseases that were collected contain noise and blurred edges, which can hinder the network’s ability to accurately extract features of the diseases. (2) The classification of disease images is a challenging task due to the high intra-class diversity and inter-class similarity of rice leaf diseases. This paper proposes the Candy algorithm, an image enhancement technique that utilizes improved Canny operator filtering (the gravitational edge detection algorithm) to emphasize the edge features of rice images and minimize the noise present in the images. Additionally, a new neural network (ICAI-V4) is designed based on the Inception-V4 backbone structure, with a coordinate attention mechanism added to enhance feature capture and overall model performance. The INCV backbone structure incorporates Inception-iv and Reduction-iv structures, with the addition of involution to enhance the network’s feature extraction capabilities from a channel perspective. This enables the network to better classify similar images of rice diseases. To address the issue of neuron death caused by the ReLU activation function and improve model robustness, Leaky ReLU is utilized. Our experiments, conducted using the 10-fold cross-validation method and 10,241 images, show that ICAI-V4 has an average classification accuracy of 95.57%. These results indicate the method’s strong performance and feasibility for rice disease classification in real-life scenarios.

## 1. Introduction

Rice is a crucial food crop globally, serving as the staple food for approximately 50% of the world’s population [1]. It holds significant importance in national agricultural production [2]. The efficient and accurate identification of rice diseases is a pressing issue for rice growers, as rice disease is the primary factor affecting both yield and quality. The loss of yield and quality caused by rice disease has a direct impact on rice production, as well as human and animal safety [3]. Given that the safe production of rice is a strategic concern related to social stability and national security, addressing this issue is of utmost importance. The traditional method of identifying rice diseases involves visual inspection by plant protection experts. However, this method is time consuming and relies heavily on subjective judgment, making it difficult to promptly prevent and control rice diseases [4]. With the rapid improvement in computer hardware processing speed and software technology, the utilization of artificial intelligence, image recognition, big data, and deep learning technology has become increasingly prevalent in the field of agriculture, particularly in crop disease diagnosis and recognition. When rice plants are affected by diseases, their physiological structure and morphological characteristics change, resulting in symptoms such as leaf discoloration, decay, and deformation. To ensure higher yield and protect rice from pests and diseases, it is crucial to develop a new method for detecting rice diseases that can quickly and accurately identify and classify them in the early stages. This will help to maintain the overall quality and yield of rice.

The agricultural sector is witnessing a shift from traditional intensive farming to emerging technologies such as precision agriculture and intelligent agriculture. Precision agriculture utilizes information technology (IT), satellite technology, geographic information systems (GISs), and remote sensing to enhance all functions and services of the agricultural sector [5]. This technology is crucial in improving crop disease and pest control and management, establishing intensive agricultural production management and operation mode, and enhancing the quality of agricultural products. Precision agriculture has begun to rely on various technologies such as mobile applications [6], smart sensors [7], drones [8], cloud computing [9], artificial intelligence (AI) [10], the Internet of Things (IoT) [11], and blockchain [12]. These technologies enable real-time processing and access to data on soil, crop, and weather conditions. Additionally, they allow for the timely detection and reporting of plant health, and provide guidance to farmers on soil management, crop maturity rotation, optimal planting time, and harvest time. The utilization of emerging technologies, such as the Internet of Things, presents vast potential for the advancement of smart and precision agriculture. These technologies allow for the acquisition of real-time environmental data, facilitating more efficient and effective farming practices. The use of IoT devices such as unmanned aerial vehicles with high spatial and temporal resolution can be beneficial for crop management. However, the adoption of smart agriculture has not been as strong as expected due to various challenges such as data acquisition and image processing methods. Khattab et al. have developed an IoT-based monitoring system specifically designed for precision agriculture applications [13]. This system provides environmental monitoring services to ensure that the crop growing environment is optimal and to predict early conditions that may lead to epidemic outbreaks. While the system is capable of obtaining real-time plant images and making predictions, the accuracy of its classification model for plant diseases remains a challenge. Yuki et al. proposed a straightforward image processing pipeline that adheres to the empirical principle of the simultaneous emergence of leaves and tillers in rice morphogenesis [14]. The team utilized an unmanned aerial vehicle to capture field images at very low flight altitudes. They then calculated the number of long blades using the proposed image processing pipeline, which included binarization, skeletonization, and tip detection. This study highlights the potential of the proposed image-based tiller counting method to aid agronomists in efficient and non-destructive field phenotypic analysis. This paper focuses on addressing the challenges faced by image processing and prediction classification models in the context of mature real-time image acquisition technology. Specifically, the paper proposes a solution to process images and classify rice diseases using acquired image data.

Currently, rice disease classification faces several challenges: (1) The collected rice leaf images often contain noise and insignificant edge details. If these images are directly used for training without proper preprocessing, the network’s ability to extract rice disease features is greatly weakened, which ultimately leads to low recognition accuracy. (2) Rice exhibits large intraclass diversity and high interclass similarity [15], evident in diseases such as blast and brown spot. Traditional convolutional neural networks struggle with accurate differentiation, thus requiring improvements in the network model to enhance its feature extraction capabilities [16].

In order to improve the accuracy of rice disease recognition, image processing is performed on collected rice disease pictures to address the issue of unclear edge details and significant noise. This step is crucial in the overall process of rice disease recognition, as it allows for a more precise and accurate identification of the disease. Image processing techniques such as image enhancement, shadow removal (edge detection), noise elimination, and feature extraction are employed in capturing disease images of rice plant leaves. These processed images are then fed into a classifier for classification [17]. In their study, Polesel et al. (2000) utilized an adaptive filter that regulates the sharpening path’s contribution. This approach allows for contrast enhancement in high-detail areas while minimizing or eliminating image sharpening in smooth regions. This is referred to as an unsharp masking technique for improving image contrast [18]. Phadikar et al. (2013) proposed a segmentation method that utilizes Fermi energy to distinguish the infected area of a rice pest image from its background. This method employs new algorithms to extract features such as the color, shape, and location of the infected site to characterize disease symptoms. To minimize information loss and reduce the complexity of the classifier, the researchers utilized rough set theory (RST) to select important features [19]. The Canny edge detection algorithm has become increasingly popular in image processing in recent years. Several studies have suggested using the Canny operator to enhance edge information in images and improve their clarity. The algorithm, proposed by John Canny in the 1980s, is a multi-level edge detection method [20]. John Canny focused on identifying the optimal edge, which required detecting edges that closely resembled the actual edge while minimizing noise interference. However, traditional algorithms face difficulty when filtering low-contrast images, as they cannot adaptively determine the filtering threshold or differentiate the target from the background. To address the aforementioned issues, Xuan et al. (2017) proposed an enhanced Canny algorithm. This algorithm utilizes the amplitude gradient histogram to derive two adaptive thresholds and connect edge points to obtain generalized chains. The algorithm calculates the average value of the generalized chains and removes those that fall below it. The image edge detection results are then obtained using a linear fitting method. Experimental results demonstrate that the improved algorithm is more robust to noise and effectively distinguishes the target from the background [21]. In their 2020 paper, Kalbasi et al. introduced an adaptive parameter selection method. This method uses the estimated noise intensity of the input image and the minimum output performance required for the application to select a value for the Canny parameter from a pre-existing configuration table, rather than calculating it at runtime [22]. This implementation of the Canny algorithm is adaptive, ensuring high edge detection performance and noise robustness in various situations. Despite this, the execution time of our proposed Canny algorithm is still lower than that of the latest cutting-edge implementation. Wang et al. (2019) proposed an improved gravitational edge detection algorithm that utilizes gravitational field intensity instead of image gradient, along with the introduction of the gravitational field intensity operator. The algorithm also includes an adaptive threshold selection method based on the mean and standard deviation of image gradient amplitude for images with varying levels of edge information. The gravitational edge detection algorithm, which is relatively simple to implement, has been shown to retain more relevant edge information and exhibit stronger anti-noise ability through empirical data analysis [23]. This paper proposes a Candy image enhancement algorithm (an improvement upon the Canny algorithm) to address the issues of noise and unclear edge details in rice leaf images. The method involves using an ideal high-pass filter and the gravitational edge detection algorithm to extract the edge mask of the rice disease image. Then, a simple detail-enhanced image is produced by combining smooth filtering and a Laplace operator. The proposed method multiplies the output by a mask and adds it to the input image to enhance edge detail. The experimental analysis demonstrates that this method significantly improves the visual effect of the image compared to the original image and homomorphic filtering. Additionally, the method effectively suppresses noise while maintaining image authenticity.

In order to solve the problem of high diversity and similarity among rice diseases, the accuracy of convolutional neural networks in rice disease image classification needs improvement. Lu et al. (2017) introduced a novel approach for identifying rice diseases using deep convolutional neural network (CNN) technology. The proposed model was evaluated using a 10-fold cross-validation strategy and achieved an impressive accuracy of 95.48%, surpassing that of traditional machine learning models. The feasibility and effectiveness of using deep convolutional neural networks to identify rice diseases and insect pests are demonstrated through simulation results of rice disease recognition. However, the success of this approach is contingent upon the availability of large-scale datasets and high-quality rice disease image samples. Additionally, selecting the optimal parameters still requires a significant number of experiments [24]. Atole et al. (2018) explored the use of deep convolutional neural networks to classify rice plants according to the health of their leaves. They implemented a three-class classifier using transfer learning from the AlexNet deep network, which was able to distinguish between normal, unhealthy, and serpentine plants. The network achieved an accuracy of 91.23% using stochastic gradient descent. These results suggest that commonly used neural networks could still be optimized for rice diseases and insect pests. However, no improvements have been made to tailor the neural network specifically for rice diseases [25]. Zhou et al. (2019) presented a rapid method for detecting rice diseases using a combination of FCM-KM and Faster-R-CNN fusion techniques. This approach not only enhances the recognition accuracy of the Faster R-CNN algorithm, but also reduces the time required for recognition. The method primarily concentrates on detecting diseases in collected images, which may not be suitable for monitoring large-scale rice cultivation. Furthermore, it also identifies diseases by monitoring the collected images [26]. Rahman et al. (2020) utilized state-of-the-art large-scale architectures, such as VGG16 and InceptionV3, which were fine-tuned to detect and identify rice pests and diseases. While the experimental results demonstrated the efficacy of these models on real datasets, they were found to be inadequate in detecting and classifying rice pests and diseases in heterogeneous backgrounds [27]. Wang et al. (2021) presented the ADSNN-BO, an attention-based deep separable Bayesian optimized neural network, as a means of detecting and classifying rice diseases from rice leaf images. The authors suggest that the use of artificial intelligence in the agricultural field can lead to rapid diagnosis and control of plant diseases. However, it is worth noting that their experiments did not utilize public datasets and as such, it may be difficult to promote their findings [28]. Haque et al. (2022) proposed a method for rice leaf disease classification and detection using YOLOv5 deep learning. The YOLOv5 model was trained and evaluated on a dataset of approximately 1500 photos. The use of a large dataset during training resulted in high accuracy [29].

While deep learning has demonstrated success in rice disease recognition, there remains room for improvement in recognition accuracy and the need to minimize model training time. To address these challenges, this paper presents the ICAI-V4 model, which offers the following main contributions:This paper proposes an algorithm for enhancing Candy images. The proposed method involves extracting the edge mask through ideal high-pass filtering and gravitational field intensity operator. The simple detail enhancement image is obtained by combining smooth filtering and a Laplace operator. The resulting image is then multiplied by the mask and added to the input image to obtain a better edge detail enhanced image.This paper proposes the ICAI-V4 model, which incorporates a coordinated attention mechanism into the backbone network. This enhancement is designed to improve the feature capture ability and overall performance of the network model. The INCV backbone structure incorporates both Inception-iv and Reduction-iv structures, while also integrating involution to enhance the network’s feature extraction capabilities from a channel perspective. This enables the network to better classify similar images of rice pests and diseases. Leaky ReLU is utilized as an alternative to the ReLU activation function to mitigate the issue of neuron death caused by the latter and enhance the model’s resilience.

## 2. Materials and Methods

### 2.1. Data Acquisition

The identification and classification of rice diseases heavily rely on data collection. The datasets used in this study were primarily obtained from open-source platforms such as Kaggle (Rice Leaf Diseases Dataset | Kaggle) and Mendeley Data (Rice Leaf Disease Image Samples—Mendeley Data). The Kaggle platform contributed 3355 rice disease images, while the Mendeley Data platform provided 5932 images. To obtain a large number of datasets for network training and address the issue of duplicated images, various techniques such as deletion, horizontal flipping, vertical flipping, and random cropping were applied after enhancing the collected images. The final result was a dataset of 10,241 images, all saved in jpg format. The distribution and proportion of different types of rice disease images can be found in Table 1. In Figure 1, the process of collecting, transmitting, enhancing, and preprocessing rice images is demonstrated. In order to avoid any noise interference, it is important to enhance the image before inputting it into the recognition network. The dataset included various types of rice diseases such as bacterial blight, blast, brown spot, and tungro disease. The images were evenly distributed and randomly divided into training and test sets at a 9:1 ratio. The training set is utilized to train the model, while the test set is used to evaluate the model’s recognition ability. To maintain the objectivity of the results, the test set remains unprocessed. However, the images in the training set are subjected to random rotations, flips, and mirroring to expand the dataset without affecting the results’ objectivity. Additionally, all images in this study were normalized to a size of 299 × 299 to expedite the model training process [30].

Our self-built dataset focuses on four primary types of rice diseases: bacterial blight, blast, brown spot, and tungro disease. Each disease is characterized by unique symptoms, which are as follows:Rice bacterial blight is caused by the bacterium Xanthomonas, which produces water-stained lesions starting from the leaf margin a few centimeters from the tip of the leaf and spreading to the leaf base. The affected area increases in length and width, and changes color from yellow to light brown due to dryness [19].Rice blast disease is characterized by the appearance of green-gray spots on infected leaves, which are surrounded by dark green borders. As the disease progresses, the spots or lesions become oval or spindle-shaped with reddish-brown borders, and some may even become rhombus-shaped. The lesions can expand and merge, ultimately leading to the complete destruction of the leaves.Rice brown spot is a fungal disease caused by Bipolaris oryzae that primarily damages the aboveground parts of rice plants, particularly the leaves. It is prevalent in all rice-growing regions, especially when there is a shortage of fertilizer and water, leading to poor growth of the rice. It often occurs in combination with rice blight. The disease causes leaf blight, resulting in reduced 1000-grain weight and an increase in empty grains, which adversely affects the yield and quality of rice. Although the harm of disease in rice production is decreasing with improved fertilization and water conservation, certain areas still experience increased incidence of ear blight in late rice due to prolonged seedling age, resulting in significant harvest losses.Rice tungro disease causes infected leaves to turn from orange to yellow, with the staining spreading from the tip to the lower part of the leaf. Infected leaves may exhibit a striped appearance and rusty spots, and may contain planthoppers [17].

This section provides a detailed explanation of image enhancement algorithms and rice disease classification models. To aid in comprehension, Table 2 presents a list of variables utilized in the following formula. Table 3 provides an explanation of the abbreviations used in this article. Figure 2 provides a flowchart of the program for this article.

### 2.2. Candy Image Enhancement Algorithm

The quality of rice disease images can be affected by various factors, such as illumination, resulting in issues such as noise and insignificant edge features. These problems can negatively impact the training and testing of the network, making identification and classification more difficult. To improve the accuracy and reliability of classification after recognition, it is necessary to enhance the image quality. This paper utilizes the Candy image enhancement algorithm to process the images. The algorithm replaces the image gradient used in the traditional Canny operator with gravitational field intensity and implements an adaptive threshold selection method based on standard deviation. The edge enhanced image is obtained by multiplying the initial enhanced image with the mask image, which is obtained through ideal high-pass filtering and gravitational field intensity operator processing. The resulting mask image is then combined with smooth filtering and the Laplacian operator to obtain a simple detail enhanced image. Finally, the edge enhanced image is obtained by adding the mask image multiplied initial enhanced image to the input image. The proposed method results in a visually improved image compared to the original and homomorphic filtering. It enhances edge details while effectively suppressing noise and preserving the authenticity of the image.

The Canny algorithm has long been used for image edge detection. However, it is known to be sensitive to noise, which often results in the loss of weak edge information while filtering out noise. Additionally, its adaptability to fixed parameters is poor. To address these issues, Weibin Rong proposed an improved algorithm based on the Canny algorithm [31]. In this paper, the algorithm proposes a new concept, gravitational field strength, to replace the traditional image gradient method. The algorithm then introduces the gravitational field strength operator and an adaptive threshold selection method based on standard deviation for images with rich edge information. By implementing these improvements, the Canny algorithm becomes simpler, more efficient in saving useful edge information, and more robust to noise [21].

The traditional Canny algorithm has the following steps:To remove noise, a Gaussian filter is applied to smooth the image. The filter is selected based on the appropriate Gaussian function to smooth the image according to the rows and columns, which is achieved by convoluting the image matrix. The Canny algorithm typically uses a two-dimensional Gaussian function (Equation (1)) to achieve this, as the convolution operation satisfies the commutative and associative laws.
(1)$$G(x,y)=\frac{e^{-\frac{x^{2}+y^{2}}{2\sigma^{2}}}}{2\pi \sigma^{2}}$$The parameter *σ* in the Gaussian filter determines the extent to which the image is smoothed, affecting its expansion.The intensity gradient of an image can be determined by analyzing the change in pixels. When there is a significant difference between adjacent pixels, it can suggest the presence of an edge in that particular area. This edge is visually represented as a transition from black to white. The first-order partial derivative in the *x* and *y* directions can be approximated using the following formula (Equations (2) and (3)): (2)$$E_{x}[i,j]=\frac{I\left[i+1,j\right]-I\left[i,j\right]+I[i+1,j+1]-I[i,j+1]}{2}$$
(3)$$E_{x}[i,j]=\frac{I\left[i,j+1\right]-I\left[i,j\right]+I[i+1,j+1]-I[i+1,j]}{2}$$

The template of the image gradient calculation operator (Equations (4) and (5)) is:(4)$$G_{x}=\left(\begin{array}{cc} -1 & 1\\ -1 & 1 \end{array}\right)$$
(5)$$G_{y}=\left(\begin{array}{cc} 1 & 1\\ -1 & -1 \end{array}\right)$$

The size and direction of the gradient can be calculated. The image gradient size (Equation (6)) is:(6)$$\left\|M(i,j)\right\|=\sqrt{E_{x}{\left[i,j\right]}^{2}+E_{y}{\left[i,j\right]}^{2}}$$

The azimuth of the image gradient (Equation (7)) is:(7)$$\theta (i,j)=arctan\frac{E_{y}[i,j]}{E_{x}[i,j]}$$

4.Non-maximum suppression is a technique used to eliminate false edge detections. It involves suppressing pixels whose gradient is not large enough and retaining only the maximum gradient, resulting in thin edges. This classical thin-edge algorithm is applied after obtaining the gradient magnitude image M[i,j] to accurately locate the edge. The Canny algorithm utilizes 3 × 3 adjacent regions, consisting of eight directions to interpolate the gradient amplitude along the gradient direction. If the magnitude M[i,j] is greater than the two interpolation results in the gradient direction, it is identified as a candidate edge point; otherwise, it is marked as a non-edge point. This process generates a candidate edge image.5.To determine the possible boundary, the double threshold method is applied. Despite non-maximum suppression, there may still be noise points in the image. Therefore, the Canny algorithm employs a technique in which a threshold upper bound and a threshold lower bound are set. The process involves marking pixels with a gradient amplitude higher than the high threshold as edge points and those with a lower gradient amplitude as non-edge points. The remaining pixels are marked as candidate edge points. Candidate edge points connected to edge points are then marked as edge points, reducing the impact of noise on the final edge image.6.The boundary is tracked using the lag technique. The candidate edge points are re-evaluated and the 8-connected domain pixels of a weak edge point are examined. If there are any strong edge points present, the weak edge point is considered to be retained as part of the edge. However, if there are no strong edge points, the weak edge is suppressed.

The traditional Canny edge detection algorithm has been widely used in practical engineering, but there are still two aspects that require improvement. Firstly, the algorithm uses the first-order finite difference 2 × 2 adjacent region to calculate the image gradient. The lack of deviation in the 45° and 135° directions may cause the loss of genuine edge information. Additionally, setting a fixed value for the double threshold in the traditional Canny algorithm can result in the loss of local feature edge information, especially in images with abundant edge information such as rice pests and diseases. Therefore, the traditional Canny algorithm may not be the most adaptable option for such images. In this paper, the authors propose the use of gravitational field strength as a replacement for image gradient. Additionally, they suggest an adaptive threshold selection method based on standard deviation. These techniques aim to improve the clarity and accuracy of image processing.

The law of universal gravitation is utilized in gravitational edge detection algorithms for image processing. In this method, each pixel is treated as an object with a mass equivalent to its gray value. However, the performance of the gravitational edge detection algorithm varies significantly in bright and dark regions. When pixels are in darker areas with lower quality, their gradients generate less total gravity compared to brighter areas. This causes the gravity method to reduce the correlation of gradient changes in dark regions, leading to a loss of edge points. Therefore, the gravitational field strength is introduced to overcome the difference between the bright and dark regions. The total gravitational field strength generated at a point in the image is a combination of the gravitational field strength generated by the surrounding pixels. In this paper, the obtained gravitational field intensity is understood as an image gradient. If the intensity on a pixel exceeds the threshold, it is classified as an edge point. The formula to calculate the gravitational field strength assigned to the point (Equation (8)) is as follows:(8)$$\underset{E_{total}}{\to }=\sum_{i=1}^{n}\frac{Gm_{i}}{{\left\|\underset{r_{i}}{\to }\right\|}^{2}}\frac{\underset{r_{i}}{\to }}{\left\|\underset{r_{i}}{\to }\right\|}$$

The gradient component in the *X* direction (Equation (9)) is:(9)$$\underset{E_{x}}{\to }=\sqrt{2}G(m\left[i+1,j+1\right]-m\left[i,j+1\right]+m[i+1,j]-m\left[i,j\right])\underset{i}{\to }$$

The gradient component in the *Y* direction (Equation (10)) is:(10)$$\underset{E_{y}}{\to }=\sqrt{2}G(m\left[i+1,j+1\right]-m\left[i+1,j\right]+m[i,j+1]-m\left[i,j\right])\underset{j}{\to }$$

Therefore, the gradient size (Equation (11)) is:(11)$$E=\sqrt{E_{x}^{2}+E_{y}^{2}}$$

The gradient azimuth (Equation (12)) is:(12)$$\theta =arctan\frac{E_{y}}{E_{x}}$$
where $\vec{i}$ and $\vec{j}$ are the unit vectors in the horizontal and vertical directions, respectively. In order to retain more edge information, the adjacent region 2 × 2 is extended to 3 × 3.

The gradient component in the *X* direction (Equation (13)) is:(13)$$\underset{E_{x}}{\to }[i,j]=G\frac{\left(m_{4}-m_{8}\right)+\sqrt{2}\left(m_{5}-m_{1}+m_{3}-m_{7}\right)}{4}\underset{i}{\to }$$

The gradient component in the *Y* direction (Equation (14)) is:(14)$$\underset{E_{y}}{\to }[i,j]=G\frac{\left(m_{2}-m_{6}\right)+\sqrt{2}\left(m_{1}-m_{5}+m_{3}-m_{7}\right)}{4}\underset{j}{\to }$$

Therefore, the gradient size (Equation (15)) is:(15)$$\left\|\underset{E}{\to }(i,j)\right\|=\sqrt{\underset{E_{x}}{\to }{\left[i,j\right]}^{2}+\underset{E_{y}}{\to }{\left[i,j\right]}^{2}}$$

The gradient azimuth (Equation (16)) is:(16)$$\theta =arctan\frac{\left\|\underset{E_{y}}{\to }[i,j]\right\|}{\left\|\underset{E_{x}}{\to }[i,j]\right\|}$$

The value of the pixel located in the upper left corner of the central pixel I[i,j] is represented by $m_{1}$. Moving clockwise around the central pixel, the values of the pixels in the adjacent positions are represented by $m_{2}$, $m_{3}$, …$m_{8}$.

The image of rice diseases and insect pests contains rich edge information and a scattered gradient amplitude distribution. However, due to the inconsistent contrast of each part of the image and the relatively large standard deviation of the image gradient, selecting a double threshold for the entire image is not sufficient for completing edge detection. This is because some edge regions with a small gradient magnitude will have a selected threshold that is too high, resulting in the loss of detailed edges. This study proposes a method for selecting double thresholds for each pixel. The first step involves calculating the average value of the gradient size of the entire image $E_{ave}$ (Equation (17)).
(17)$$E_{ave}=\sum_{i=1}^{m}\sum_{j=1}^{n}\frac{\left|E[i,j]\right|}{m\times n}$$

If the gradient size of the pixel I[i,j] is less than 15–20% of the average gradient amplitude ($E_{ave}$), it will be classified as a non-edge point. This step is crucial in preventing the algorithm from introducing more noise in areas with fewer edges, such as those with small average gradient amplitude and standard deviation, in rice pest images. To determine the threshold of pixel I[i,j], the average and standard deviation of the gradient sizes of an N × N matrix are calculated. N is an odd number and is typically greater than 20. The center of the threshold is the pixel I[i,j]. The pixel threshold (Equation (18)) can then be obtained by:(18)$$T_{h}=E_{ave}+k\sigma$$

To obtain the threshold for each pixel in the boundary region of an image with a matrix less than N × N, the insufficient part is first set to empty. The mean and standard deviation of the matrix are then calculated. This process results in each pixel having its corresponding double threshold, which can be used to detect and connect the edges of the entire image [32].

The Candy image enhancement algorithm follows a specific process, which is illustrated in Figure 3:The first step in the list number image processing pipeline involves preprocessing the original image. This includes normalization and conversion to grayscale if the original image is in color. The resulting normalized and grayscale image is then used as input for the subsequent steps.The process of extracting the mask and detail enhancement images involves two steps. Firstly, the input image is processed using the improved canny algorithm to extract the edge details. The detected image edge information is then utilized as the mask. In the second step, the input image undergoes a process of smoothing and denoising. To deal with the noise, the Laplace operator is used, which can have a strong response to the noise and cause negative effects. Therefore, denoising processing should be carried out prior to using the Laplacian operator. This allows the denoised image to better highlight small details in the image.A preliminary detail-enhanced image can be obtained by adding the result of the Laplacian operator to the input image. However, since the Laplacian operator is isotropic, it detects isolated points effectively, but may cause loss of the edges of the square. Therefore, further operations are required to preserve the information.To extract the image well, the processed mask image is multiplied with the preliminary enhanced image. The image enhancement algorithm results from adding the edge and detail information to the input image [33].

The rice leaf image was processed using the Candy image enhancement algorithm, as depicted in Figure 4.

### 2.3. Identification of Rice Diseases Based on ICAI-V4 Model

In the field of rice pest and disease identification, leaf images exhibit significant intra-class diversity and inter-class similarity. To overcome this challenge, it is crucial to enhance the feature extraction capability of convolutional neural networks. This can be achieved by obtaining more detailed features and reducing the false positive rates. Deep convolutional networks have been instrumental in significantly improving image recognition performance in recent years and are therefore considered the core of this advancement. The Inception-V4 architecture has demonstrated impressive performance while requiring relatively low computational resources. This paper selects the Inception-V4 model as the primary component of the neural network framework. To ensure that the entire model can be stored in memory, the model was divided into six subnetworks and trained accordingly. The Inception architecture offers a high level of tunability, allowing for numerous adjustments in the number of filters per layer without compromising the quality of the fully trained network. Each subnetwork is inspired by traditional image filtering techniques and boasts two key features that contribute to its appeal and widespread use: spatial agnosticity and channel specificity. The convolution kernel’s ability to adapt to different visual modes in different spatial locations is limited, which raises questions about its flexibility for different channels.

To address the limitations mentioned above, this paper proposes a modification to the Inception blocks by incorporating involution to the subnetwork structure. This operation possesses symmetrical inverse inherent characteristics, which enhance flexibility across different channels and improve the network’s learning effect compared to convolution [34]. To address the issue of neuron death caused by the ReLU function, the Leaky ReLU activation function is utilized. Furthermore, coordination concerns are integrated into the backbone network to enhance the overall performance. The proposed method has the ability to capture not only cross-channel information, but also direction-aware and location-sensitive information. This feature helps the model to identify and locate objects of interest more accurately and enhance features by emphasizing information representation.

The ICAI-V4 network structure boasts excellent feature extraction capabilities and robustness. This results in improved accuracy in recognizing and classifying similar disease features, while also reducing the model’s running time. The structure comprises a backbone module, a coordination attention module, an INCV structure, an average pooling layer, a discard layer (with a keep probability of 0.8), and a classification block. The network structure of ICAI-V4 is illustrated in Figure 5. The primary workflow can be summarized as follows:The initial layer of ICAI-V4 is the Stem layer, which primarily serves to rapidly decrease the resolution of the feature map. This reduction in resolution allows for subsequent inceptions to decrease the amount of computation required. Additionally, the activation function of the standard convolution layer has been modified from ReLU to Leaky ReLU, which enhances the network’s robustness.The second module in ICAI-V4 is the coordination attention module. This module is capable of capturing cross-channel information, as well as direction-aware and position-sensitive information. This enables the model to more accurately locate and identify objects of interest and enhance features by emphasizing information representation.The third component of ICAI-V4 is the INCV structure, which consists of 1 × 1 convolution, 3 × 3 convolution, pooling layer, and asymmetric convolution decomposition. The involution layer is incorporated into various structures to enhance the flexibility of different channels and improve the overall network learning effect. The network comprises three Inception-iv structures and two Reduction-iv structures used multiple times. Refer to Figure 5 for a detailed illustration of the specific structure.ICAI-V4’s fourth component consists of an adaptive pooling layer, data dimensionality reduction, linear layer, and a discard layer. The classification results of input rice pest images are determined by a softmax activation function.

#### 2.3.1. Coordinate Attention

In order to enhance the effectiveness of the model’s performance, this paper proposes the addition of a coordinated attention mechanism to the backbone network of Inception-V4. The attention mechanism is divided into two modules: channel attention and spatial attention. While channel attention has proven to be effective in improving model performance, it often disregards location information. In their recent study, Hou et al. (2021) introduced a novel approach to mobile network attention mechanisms, termed ‘coordinate attention’. This method involves incorporating location data into channel attention, resulting in improved network performance [35].

In coordinating attention using two 1D global pooling operations, input features along the vertical and horizontal directions are aggregated into two separate directional software feature maps. These feature maps, which contain embedded direction-specific information, are then encoded into two attention graphs. Each attention graph captures the long-distance correlation of the input feature map along a spatial direction. The proposed attention method, named coordinate attention, can save location information in the generated attention map [36]. This allows for the application of two attention maps to the input feature map through multiplication, resulting in an emphasized representation of interest. The operation of this method distinguishes spatial directions and generates a coordinate-aware attention map. Coordinated attention not only captures cross-channel information, but also direction-aware and position-sensitive information. This additional information aids the model in accurately identifying and locating objects of interest [37].

A coordinate attention block serves as a computing unit that amplifies feature expression in mobile networks. It takes an intermediate characteristic tensor $X=[x_{1},x_{2},\ldots ,x_{c}]\in R^{C\times H\times W}$ as input and outputs $Y=[y_{1},y_{2},\ldots ,y_{c}]$ with the same size and enhanced representation, achieved through transformation.

The global pooling method is often utilized for encoding channel attention and spatial information. However, it can be challenging to retain location information since it compresses global spatial information into channel descriptors. To address this issue, the coordinate attention method decomposes global pooling into one-to-one feature encoding operations (Equation (19)). This allows the attention module to capture remote spatial interactions with accurate location information [38].
(19)$$Z_{c}=\frac{1}{H\times W}\sum_{i=1}^{H}\sum_{j=1}^{W}x_{c}(i,j)$$

In order to encode each channel in input *X*, a pooled kernel of size (*H*, 1) or (1, *W*) is utilized along the horizontal and vertical coordinates. This results in the output of channel c with height h being expressed as (Equation (20)):(20)$$Z_{c}^{h}(h)=\frac{1}{W}\sum_{0\leq i\leq W}x_{c}(h,i)$$

Similarly, the output of channel c with width w can be written as (Equation (21)):(21)$$Z_{c}^{w}(w)=\frac{1}{H}\sum_{0\leq j\leq H}x_{c}(j,w)$$

The above two transformations generate feature maps along two spatial directions, enabling the attention module to capture long-term dependencies along one direction and preserve accurate location information along another. This aids the network in more precise target localization.

After the information embedding transformation, the resulting output is concatenated with another transformation, and then passed through the convolution transformation function (Equations (22)–(24)).
(22)$$f=\delta (F_{1}(\left[z^{h},z^{w}\right]))$$
(23)$$g^{h}=\sigma (F_{h}(f^{h}))$$
(24)$$g^{w}=\sigma (F_{w}(f^{w}))$$

Finally, the output *Y* of the coordinate attention block (Equation (25)) can be written as:(25)$$g_{c}(i,j)=x_{c}(i,j)\times g_{c}^{h}(i)\times g_{c}^{w}(j)$$

The specific structure of coordinate attention, which is added to the backbone network in this paper, aims to improve the model’s performance and feature grasping ability. This can be seen in Figure 5.

#### 2.3.2. INCV Blocks

The high intra-class diversity and inter-class similarity of rice leaf diseases require the network model to effectively extract feature information, thus improving the accuracy and efficiency of the model. Despite the rapid development of neural network architecture, convolution remains the backbone of deep neural network construction. The convolution kernel, inspired by classical image filtering methods, has two significant characteristics that contribute to its popularity and effectiveness: spatial agnosticity and channel specificity. While the concepts of spatial agnosticism and spatial compactness may seem beneficial in terms of increasing efficiency and explaining translation equivalence, they may limit the adaptability of convolution kernels to varying visual modes in different spatial locations. This raises concerns about the ability of convolution kernels to flexibly adjust to different channels.

In their recent paper, Li et al. (2021) introduced involution as a potential solution to address the limitations of convolution [34]. Involution offers symmetrical inverse characteristics that are space specific and channel-unknown. This means that the involution cores are unique to each spatial location, but shared across channels. If the involution kernel is parameterized into a fixed-size matrix, such as a convolution kernel, and updated using a back-propagation algorithm, it may not be able to transmit between input images with varying resolutions due to its space-specific properties. To address variable feature resolution, the involution kernel for a specific spatial position can only be generated when the corresponding position’s incoming feature vector is present. This approach is intuitive and effective. Additionally, Involution reduces core redundancy by sharing involution cores along the channel dimension. The computational complexity of the involution operation increases in a linear fashion as the number of feature channels increase. This means that the dynamic parametric involution kernel is capable of covering a wide range of spatial dimensions. Reverse design allows for two advantages with involution: (i) it can summarize context in a wider spatial arrangement, overcoming the challenge of effectively modeling long-distance interaction; (ii) involution can assign adaptive weights to different locations to prioritize visual elements with the most information in the spatial domain [39].

In contrast to convolution, involution involves sharing the kernel in the channel dimension while utilizing a space-specific kernel for more adaptable modeling in the spatial dimension. The size of the involution kernel (Equation (26)) is:(26)$$H_{I}\in R^{H\times W\times K\times K\times G}$$
where *H* and *W* denote their height and width, *K* is kernel size, and G<C denotes that all channels share G kernels. In the context of involvements, the fixed weight matrix is not utilized as a learnable parameter, unlike in convolution. Instead, the involvements kernel is generated based on the input feature mapping to ensure the automatic alignment of kernel size and input feature size in the spatial dimension [40]. Unlike the convolution kernel, the shape of the involution kernel $H_{i}$ depends on the shape of the input feature map. A natural idea is to generate an involution kernel based on the original input tensor, so that the output kernel and the input are easily aligned. We express the kernel generating function as ∅, and abstract the function mapping at each position (i,j) (Equation (27)) as [34]: (27)$$H_{i,j}=\emptyset (X_{\varphi_{i,j}})$$
where $\varphi_{i,j}$ is an index set of the neighborhood of coordinates (i,j), and $X_{\varphi_{i,j}}$ represents a patch containing $X_{i,j}$ on the feature map.

The output feature map of the involution is obtained by performing a multiplicative addition operation on the input with such an involution kernel (Equation (28)). The definition is as follows:(28)$$Y_{i,j,k}=\sum_{(u,v)\in ∆k}H_{i,j,u+\frac{K}{2},v+\frac{K}{2},\frac{kG}{c}}X_{i+u,j+v,k}$$

Figure 6 displays the complete involution diagram, depicting the process of generating the involution convolution kernel and subsequently producing a new feature diagram. In this method, the representation of a pixel of size 1 × 1 × C is reduced to C/r through linear change, where *r* represents the reduction ratio. This is then changed to K × K × G, which is involution kernels. When *G* is 1, the K×K×1 kernel is multiplied with the pixels in the K × K field of the current pixel in the original feature map, and repeated on *C* channels to obtain the three-dimensional matrix of K × K × C. The width and height of the three-dimensional matrix are summed in two dimensions, while the channel dimension is retained. This generates a 1 × 1 × C vector in the sex feature diagram, where the value of *C* channels in a pixel position is generated at once.

The involution kernel $H_{i,j}\in R_{v}$ is created using a function *φ* that takes a single pixel at (*i*, *j*) as a condition and rearranges the channel to space. The multiplication and addition operations of the involution are divided into two steps. The product operation multiplies the tensors of C channels with the involution kernel H, respectively, while the addition operation adds the elements within the involution kernel to the involution kernel.

Involution has the following advantages over convolution:Sharing kernels on the channel improves performance by allowing us to use large spatial spans while maintaining design efficiency through the channel dimension. This is true even if weights are not shared in different spatial locations, as it does not significantly increase the number of parameters and calculations.While the kernel parameters of each pixel in the space are not directly shared, involution does share meta-weights at a higher level, specifically the parameters of the kernel generating function. This allows us to still share and migrate knowledge across different spatial locations. However, freeing the limitation of the convolution sharing kernel in space and allowing each pixel to learn its corresponding kernel parameters does not necessarily lead to better results, despite addressing the issue of large parameter increase.

The Inception-V4 model is trained in a partitioned manner, consisting of six subnetworks. The Inception architecture is highly flexible, with each subnetwork incorporating multiple branches inspired by classical image filtering methods. These branches exhibit spatial agnosticity and channel specificity, allowing for effective feature extraction. The convolution kernel’s ability to adapt to different visual modes in different spatial locations is limited, which questions its flexibility for different channels. Involution, on the other hand, is capable of modeling long-distance interactions and is sensitive to channel information. This paper proposes enhancements to the inception and reduction structures in the backbone network infrastructure of Inception-V4. The subnetworks now include a 1 × 1 convolution and an involution layer, which enhances the network’s ability to extract features from channel perspectives. As a result, it can better classify similar images of rice pests and diseases. To better fit the Inception architecture, a 1 × 1 convolution is added to adjust the input and output parameters. After adding this convolution and Involution layer to the structure, the model’s accuracy is shown, as in Table 4. The results demonstrate that the proposed INCV structure has higher accuracy and is more suitable for rice pest classification [40].

#### 2.3.3. Leaky ReLU

The ReLU function is a commonly used activation function in deep learning. Its expression is (Equation (29)):(29)$$\mathrm{ReLU}=\left\{\begin{array}{c} 0,x<0\\ x,x\geq 0 \end{array}\right.$$

The ReLU function has the following characteristics:When the input is positive, there is no gradient saturation problem.The calculation speed is much faster. There is only a linear relationship in the ReLU function, so its calculation speed is faster than the Sigmoid function and the tanh function [41].The Dead ReLU problem arises when the input to the ReLU activation function is negative, rendering it completely invalid. This is not a concern during forward propagation as only certain areas are sensitive to negative input. However, during back propagation, if a negative number is encountered, the gradient will be zero, resulting in non-convergence of calculation results and neuronal death. This prevents weight updates, leading to the problem of non-updating neurons.The output of the ReLU function is zero or positive, which means that the ReLU function is not a zero-centered function.

To address the Dead ReLU problem associated with the ReLU activation function, this paper proposes the use of Leaky ReLU. The proposed activation function is depicted in the accompanying figure. The expression for Leaky ReLU (Equation (30)) is:(30)$$f(x)=\left\{\begin{array}{c} x,x\geq 0\\ ax,x<0 \end{array}\right.$$

The parameter ‘a’ is randomly generated during model training and eventually converges to a constant value. This process enables one-sided suppression while retaining some negative gradient information to prevent complete loss.

The Leaky ReLU function has the following advantages:Some neurons will not die. The Dead ReLU problem of the ReLU function is solved;The linear and unsaturated form of Leaky ReLU can converge quickly in SGD;The Leaky ReLU function has a faster calculation speed compared to the sigmoid and tanh functions. This is because it has a linear relationship and does not require exponential calculations. Nonlinearity typically requires more computation and thus results in slower execution speeds.

### 2.4. Evaluating Indicator

This paper examines the classification of rice diseases and evaluates model performance using indicators such as accuracy, precision, recall rate, F1 score, and ROC/AUC.

Assuming that all images categorized under rice blast disease are positive samples, the remaining disease images are considered negative samples and do not belong to the rice blast disease category. The study focused on the category of rice blast and used it as an example. TP, which stands for true positive, refers to rice images that were accurately classified as having rice blast disease and were placed in the correct category. FP (false positive) refers to incorrectly predicting a rice image with other diseases as rice blast disease. TN (true negative) refers to predicting an image of rice with other diseases as belonging to their respective categories. FN (false negative) refers to mistakenly predicting an image of rice with rice blast disease as belonging to another disease category. Table 5 displays the structure of the confusion matrix using a binary task as an example.

Accuracy (Equation (31)) is defined as the ratio of correctly classified samples to the total number of samples, expressed as a percentage. It is a commonly used metric to evaluate the performance of a classification model:(31)$$Accuracy=\frac{TP+TN}{TP+TN+FP+FN}$$

The accuracy rate (Equation (32)) is a metric that measures the proportion of correctly predicted positive samples out of all of the samples that are predicted to be positive by the model:(32)$$Precision=\frac{TP}{TP+FP}$$

Recall rate (Equation (33)), also known as sensitivity or true positive rate, is a measure of the proportion of actual positive samples that are correctly identified as positive by a prediction model. In other words, it is the ratio of true positives to the sum of true positives and false negatives [30]:(33)$$Recall=\frac{TP}{TP+FN}$$

The *F*1 score (Equation (34)), also known as the *F*-score or *F*-measure, is a measure of a test’s accuracy. It is calculated as a weighted average of the precision and recall, which are measures of the test’s ability to correctly identify positive results and correctly exclude negative results, respectively. The formula for calculating the *F*1 score is:(34)$$F1=\frac{2\times precision\times recall}{precision+recall}$$

## 3. Results

### 3.1. Experimental Environment and Data Preparation

Hardware: Intel (R) Core (TM) i7-10875H CPU @ 2.30GHz; graphics card: NVIDIA GeForce RTX 2060; system memory: 16.0 GB. Software environment: Anaconda Nayigator 2.1.2; PyTorch 1.10; Python 3.8.12; MATLAB R2021b. Operating system: Windows 10.

The training of machine learning models for detecting rice diseases requires a large number of disease samples. However, obtaining sufficient disease images is a challenging task. To overcome this challenge, it is necessary to expand the dataset to increase the number of samples before training the model. This approach can help prevent the overfitting of the network and improve the model’s performance by generating equivalent values for limited data. To increase the quantity of images, geometric transformations such as horizontal flipping, vertical flipping, and random cropping can be applied. The direction of the image does not affect the classification of rice diseases. Therefore, simulating the shooting of rice images from different angles through geometric transformations can help restore the real shooting situation.

### 3.2. Experimental Design and Results

The experimental dataset in this paper includes four types of rice diseases: bacterial blight, blast, brown spot, and tungro. To ensure consistency, the input image size was standardized to 299 × 299 × 3. Following image enhancement and data expansion, a total of 10,241 rice disease images were obtained. To ensure the reliability and accuracy of the model, K-fold cross-validation was employed to train and test the model in all subsequent verification experiments. Cross-validation is a technique used to assess the performance of a model. It involves dividing the original data into groups, where most of the samples are used as the training set to train the model, and the remaining samples are used as the test set to evaluate the model’s performance. The test error of the small samples is determined and their sum of squares is recorded. This process is repeated until all samples have been predicted and tested once. This article employs a K value of 10 for 10-fold cross validation. The rice disease dataset was split into 10 groups, with 9 groups being used for training and 1 group for testing. The accuracy of the model was measured by averaging the results from the 10 tests. To enhance the diversity of the dataset partitioning and improve the model’s reliability and accuracy, we employed 10-fold cross-validation. The dataset was randomly divided 10 times, and each model was trained and tested accordingly. The overall classification accuracy of the network model is determined by retaining the best results of each 10-fold cross-validation and using the average of these 10 results. During the process of network training, it is crucial to adjust the hyperparameters appropriately. One commonly used method for hyperparameter adjustment is grid search, which helps in selecting the optimal set of hyperparameters. Grid search involves optimizing the model by traversing different hyperparameter combinations. When training a network model, it is important to pay attention to and adjust the following hyperparameters: learning rate, batch size, epoch, and momentum parameter. The paper describes the experimental setup for a machine learning model, including the values used for the learning rate (0.1, 0.01, 0.001), batch size (16, 32, 64), epoch (10, 20, 30), and momentum parameter (0.2, 0.4, 0.9). To ensure reliable results in the grid search process, a three-fold cross validation approach is employed. The study determined that the optimal hyper-parameter combination is achieved with a learning rate of 0.001, a batch size of 32, an epoch of 20, and a momentum parameter of 0.9. Further grid search was conducted on the Adam, RMSprop, and SGDM optimizers, and the RMSprop optimizer was found to perform the best under the aforementioned hyperparameter combination. The final hyperparameter combination chosen is presented in Table 6. To ensure accuracy and reliability, all experiments were conducted in the same hardware and software environment, and all models were trained using identical hyperparameter settings. The training results of the 10-fold cross-validation are depicted in Figure 7.

### 3.3. Verifying the Effectiveness of Data Preprocessing

To ensure the reliability of the rice disease datasets, similar data extensions were performed on unenhanced datasets processed by the Candy image enhancement algorithm after image preprocessing, including image enhancement and data extension. This involved utilizing methods such as horizontal and vertical flips, as well as random clipping. After applying the Candy operator filtering, four distinct datasets were obtained: the original dataset without any enhancements, the initial dataset with improved Candy operator filtering, the extended dataset without any enhancements, and the extended dataset with improved Candy operator filtering. After inputting the datasets into the proposed ICAI-V4 model for training and testing, the results in Table 7 show that data expansion can improve accuracy regardless of whether image enhancement is performed. This is due to the smaller sample size of the original dataset, and data expansion provides more samples for the model, increasing the diversity of samples and improving the model’s generalization ability.

### 3.4. Ablation Experiment

To assess the effectiveness of the method, an ablation experiment was conducted. The ablation experiments use the same experimental setting and a 10-fold cross-validation method using augmented datasets after adding noise to some samples. This paper utilized InceptionV4, AlexNet, and ResNeXt as general frameworks for the network, and conducted ablation experiments using one or more of the methods proposed in the paper. This study aims to validate the efficacy of Coordinate Attention, INCV, and Leaky ReLU through experimentation. The results of the experiment are presented in Table 8.

Table 8 demonstrates that Coordinate Attention, Involution, and Leaky ReLU all contribute to enhancing the classification accuracy of rice. Notably, ICAI-V4 exhibits the highest average classification accuracy of 95.57%. In order to enhance the efficiency of the model, the coordinated attention mechanism is utilized to transform the internal location information into channel attention. This enhances the network’s ability to extract detailed features of rice diseases. To increase the flexibility of the convolution kernel and optimize performance across different channels, we propose an INCV architecture that employs dynamic weight assignment to prioritize the most informative visual elements in the spatial domain. Our results demonstrate that the addition of both INCV and Coordinate Attention to the basic neural network leads to higher accuracy, as shown in the table below. When utilizing both Coordinate Attention and Leaky ReLU in the AlexNet network, the accuracy reaches its peak at 94.13%. The integration of Coordinate Attention, INCV, and Leaky ReLU led to improved accuracy in all three models. Through analyzing the ablation experimental results, it has been found that ICAI-V4 can achieve higher classification accuracy and exhibit relatively strong classification performance and practical value.

### 3.5. Comparison with Other Networks

In the realm of digital photography, the images captured are often subject to noise interference due to the internal characteristics of the equipment and the imaging environment. This interference can manifest as composite noise, which is a combination of Gaussian noise and impulse noise. All models listed in Table 9 underwent retesting to simulate compound noise. The standard deviation of Gaussian noise was set to 15–60, while the ratio of impulse noise was set to 0.02–0.08. To test the model’s anti-interference ability, synthetic noise can be added to the sample and then removed to restore the image to a more realistic quality. In order to validate the proposed rice disease classification model, multiple models, including the ICAI-V4 model as well as classical and recent models, were tested such as MobileNetV3 (Tarek et al., 2022) [42], EfficientNet (Chen et al., 2021) [43], DenseNet121 (Huang et al., 2017) [44], and EfficitentNetV2 (Sunil et al., 2022) [45]. This approach allowed us to establish the reliability and accuracy of our model in comparison to others. AlexNet is an early deep convolutional neural network, while ResNet50 proposes a residual block that allows for cross-layer connections, enabling features of different layers to transmit to each other and alleviating the problem of gradient disappearance. ResNeXt combines the advantages of ResNet and Inception by using group convolution, residual structure, and feature layer connections. MobileNetV3 is a lightweight network that combines the benefits of MobileNetV1 and MobileNetV2. On the other hand, efficientNetV1 uses neural architecture search technology and introduces composite coefficients to expand the network from three dimensions of network depth, width, and image resolution. EfficiencyNetV2 introduces an enhanced progressive learning approach built on the foundation of EfficienceNetV1. This updated model boasts faster training speeds and fewer parameters. In summary, the structure of this model is unique and it offers distinct advantages in image classification tasks. As such, it is essential to compare our model with others in order to demonstrate the validity and reliability of our proposed approach.

According to Table 9, Inception V4 has the lowest average classification accuracy for rice diseases compared to all other models. However, our proposed ICAI-V4 model has a good classification effect with an accuracy of 95.57%. Additionally, when random compound noise of Gaussian noise and impulse noise were added to the dataset, the average classification accuracy of ICAI-V4 only decreased by 0.32%. After adding noise, the average accuracy reached its highest in the test set.

### 3.6. Classification Performance Evaluation of ICAI-V4 and Other Networks

In this study, a larger dataset with added noise was utilized to demonstrate the effectiveness of Inception IV4 in accurately classifying rice diseases. The results show the validity of using Inception IV4 for this task. The evaluation of the model is carried out through the use of a confusion matrix, as illustrated in Figure 8, which provides a clear representation of the classification results for each rice disease by various models. In Table 10, the ICAI-V4 model’s classification performance evaluation is presented. The paper evaluated the average accuracy, recall rate, and *F*1 scores of four rice diseases. The results showed that both ICAI-V4 and DenseNet121, proposed in this paper, achieved impressive results in classifying these diseases. The average *F*1 score was approximately 95%. The classification effectiveness of blast and brown spot in most models is generally low due to the similarities in their disease characteristics. This similarity makes it challenging to train the models and places higher demands on their ability to extract feature information. Compared to other models, the ICAI-V4 proposed in this paper exhibits exceptional classification performance. Specifically, the F1 average scores for bacterial blight, blast, brown spot, and tungro are 97.54%, 93.38%, 94.98%, and 97.34%, respectively. Furthermore, our model demonstrated a slight improvement in the classification accuracy of blast and brown spot. The addition of a coordinated attention mechanism to the backbone structure of ICAI-V4 allows for the utilization of feature information between channels, resulting in an improved ability to extract image information. This enhancement is particularly beneficial when dealing with the classification of similar diseases.

### 3.7. Formatting of Mathematical Components

To assess the effectiveness of ICAI-V4, it was retrained and tested on three different datasets: PlantVillage, Stanford Automobile, and ImageNetDogs. The specifics of each dataset, such as the number of classes, total number of samples, and source, are detailed in Table 11. The proposed model was evaluated using these public datasets, and the results of the tests are presented in Table 12. Our proposed model demonstrated strong performance on three publicly available datasets, suggesting that it possesses a high degree of generalizability to other datasets.

## 4. Discussion

The effectiveness of ICAI-V4 in identifying rice leaf diseases was assessed and compared to other well-known networks including AlexNet [46], ResNet-50 [47], ResNeXt-50 [48], Inception-V4 [49], MobileNet-V2 [50], and DenseNet-121 [44]. The results presented in Table 10 demonstrate that ICAI-V4 outperformed the traditional mainstream networks in terms of recognition rate for the four rice diseases in the dataset. AlexNet employs dropout regularization to mitigate overfitting in its fully connected layers. To preserve the most salient features, maximum pooling is preferred over average pooling to avoid blurring effects. Furthermore, AlexNet suggests using a stride length smaller than the pooling kernel size to achieve overlap and coverage between the pooling layer outputs, thereby enhancing feature richness and minimizing information loss. The calculation process is straightforward and the convergence speed is rapid. However, the recognition accuracy is limited to 93.41%, 88.4%, 89.16%, and 87.05%. The ResNet-50 deep residual network addresses the issues of low learning efficiency and ineffective accuracy improvement caused by increasing network depth to some extent. While using a deep network layer can be beneficial, it can also result in the selective discarding of some layers, leading to information blocking and decreased recognition accuracy. In this particular case, the recognition accuracy was measured at 86.49%, 81.29%, 85.05%, and 84.19%. ResNeXt-50, a modification of ResNet-50, widened the network and improved its robustness and identification accuracy (91.84%, 84.95%, 89.20%, 89.67%). The network achieved better identification results for four rice leaf diseases compared to ResNet-50 (+5.35%, +3.66%, +4.15%, +5.48%). However, this improvement came at the cost of a significant increase in parameters and a decline in recognition efficiency. Inception-V4 is designed with a sparse network architecture, allowing it to perform well even with limited memory and computational resources. Its performance is impressive, achieving up to 81.63%, 77.97%, 82.51%, and 82.56%. On the other hand, MobileNet-V2 is a network that utilizes deeply separable convolution and shortcuts. The model utilizes Inverted Residuals and Linear Bottlenecks, resulting in a fast training process and a low number of parameters. While the network layer is not as deep, its recognition accuracy is lower (93.07%, 91.06%, 90.99%, 89.55%). On the other hand, DenseNet-121 employs a dense connection structure that imports the output of each layer into all subsequent layers, effectively reducing the number of parameters in the network. The use of a dense connection structure can effectively address the issues of gradient disappearance and parameter sparsity, resulting in improved generalization ability and model accuracy. In fact, the model achieved an accuracy of (95.01%, 91.73%, 91.50%, 95.18%). The study found that while MobileNet-v2 and Densenet121 had higher recognition accuracy for rice diseases, the ICAI-v4 proposed in the paper achieved even higher accuracy rates of (97.54%, 93.38%, 94.98%, 97.34%) in rice disease classification. The paper provides a summary of the reasons why ICAI-V4 is superior to other mainstream networks.

This study utilized the Candy algorithm to enhance images prior to input classification. The results, as shown in Figure 3, demonstrate that this method significantly improves the visual quality of images compared to traditional filtering methods. The algorithm enhances edge details while suppressing noise, which maintains the authenticity of the image. The results of Table 7 indicate that the classification accuracy of the model for rice diseases improved by 5.21% when using this method compared to no image enhancement.This academic paper introduces a coordinated attention mechanism into the backbone network to enhance the feature capture capability and overall performance of the network model. The experiment in Table 8 shows that the classification accuracy of Inception-V4, AlexNet, and ResNeXt for rice diseases improved after the addition of the coordinated attention mechanism (+7.79%, +1.57%, +2.14%).The INCV trunk structure is a combination of Inception-iv and Reduction-iv structures, with the addition of involution to enable more flexible modeling in spatial dimensions. This approach also incorporates a channel dimension shared kernel with a space-specific kernel to enhance the feature extraction capability of the entire network. As a result, the classification of similar rice disease images is improved. This paper incorporates involution into Inception-V4 and conducts experiments by adding involution to each part. The results, presented in Table 4, demonstrate that this matching enhances the network’s feature extraction ability and improves its classification accuracy (81.27%, 82.51%, 82.93%, 85.14%, 85.94%, 87.74%).The use of Leaky ReLU as a replacement for the ReLU activation function has been shown to alleviate the problem of neuron death caused by the latter, while also enhancing the model’s elasticity. In fact, Table 8 demonstrates that changing the activation function to Leaky ReLU resulted in a 2.87% improvement in the classification accuracy of rice diseases.According to our research, this method has shown superior classification accuracy compared to other deep neural network models. The average accuracy is 95.57%, with a recall rate of 95.54% and an F1-score of 95.81%.

Although existing methods for detecting rice disease are effective, they may not be as accurate when the diseased leaves are in the early stages and their characteristics have not fully developed. Additionally, the automatic detection of rice disease has not been widely implemented and real-time dynamic detection has not been thoroughly researched. Current methods mainly focus on detecting disease in collected images. Further research is required to determine how this method can be applied to the large-scale detection of rice planting and disease dynamics. For the future, in the process of promoting this method, it is necessary to combine intelligent Internet facilities such as agricultural Internet of Things and mobile terminal processors to realize the real-time monitoring of grain storage and pest identification, which is conducive to promoting the modernization and intelligence of the agricultural industry.

## 5. Conclusions

This paper presents a method for recognizing and classifying rice diseases using an improved Canny algorithm and ICAI-V4 model. The proposed method first utilizes an image enhancement algorithm based on the improved Canny algorithm to highlight edge details and reduce image noise. The enhanced image is preprocessed using various techniques sch as horizontal flipping, vertical flipping, and random cropping to increase the number of datasets. This extended dataset is then fed into the ICAI-V4 network model. The network model is improved by introducing a coordinated attention mechanism and designing an INCV structure, which enhances the ability of the network to extract feature information from rice disease images. The study found that the implementation of the 10-fold cross-validation method resulted in an average classification accuracy of 95.57% for ICAI-V4 when dealing with datasets containing random composite noise. These results demonstrated that the image recognition and classification method for rice disease, which is based on the improved Canny algorithm and ICAI-V4 model, has a strong ability to resist interference and is feasible for real-world applications.

The identification and classification of plant diseases using image recognition and classification technology remains a crucial research area for improving the agricultural economy. Despite the remarkable achievements made by deep learning, there is still a need for further improvement in this field. The number of rice disease samples collected in this study is insufficient. To improve the model’s generalization ability and better serve the needs of agricultural development, it is necessary to expand the dataset and increase the testing scale.

## Figures and Tables

**Figure 1 plants-12-02225-f001:**
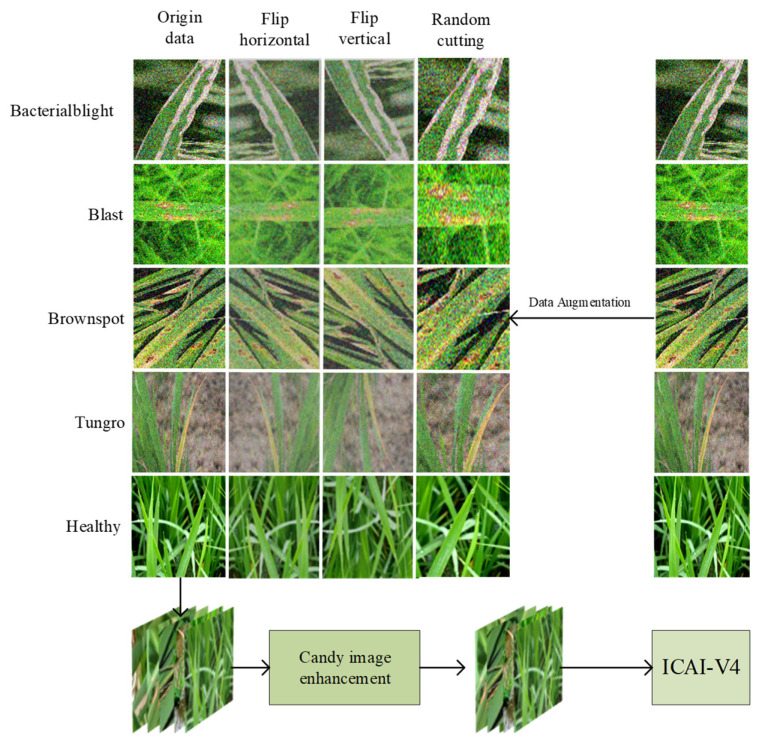
Process of acquisition, transmission, enhancement (flip, crop), and preprocessing of four rice disease images.

**Figure 2 plants-12-02225-f002:**
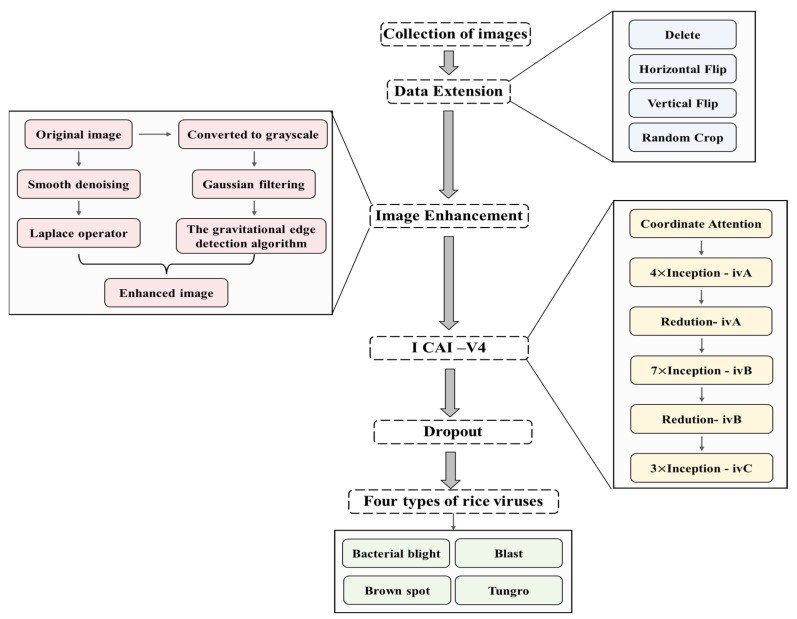
Main flow chart of rice disease classification.

**Figure 3 plants-12-02225-f003:**
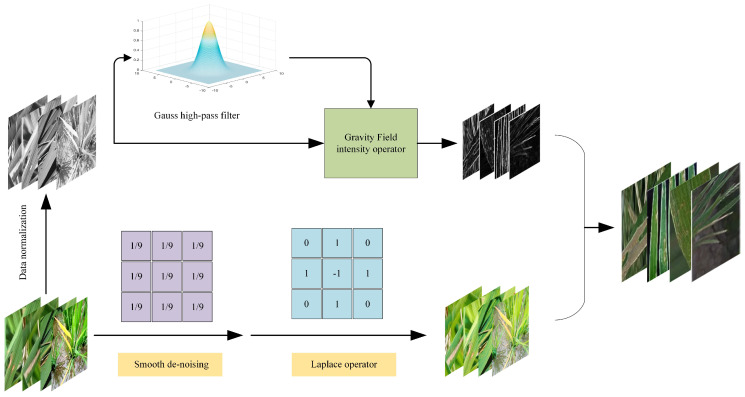
Main process of rice leaf image processing with Candy algorithm.

**Figure 4 plants-12-02225-f004:**
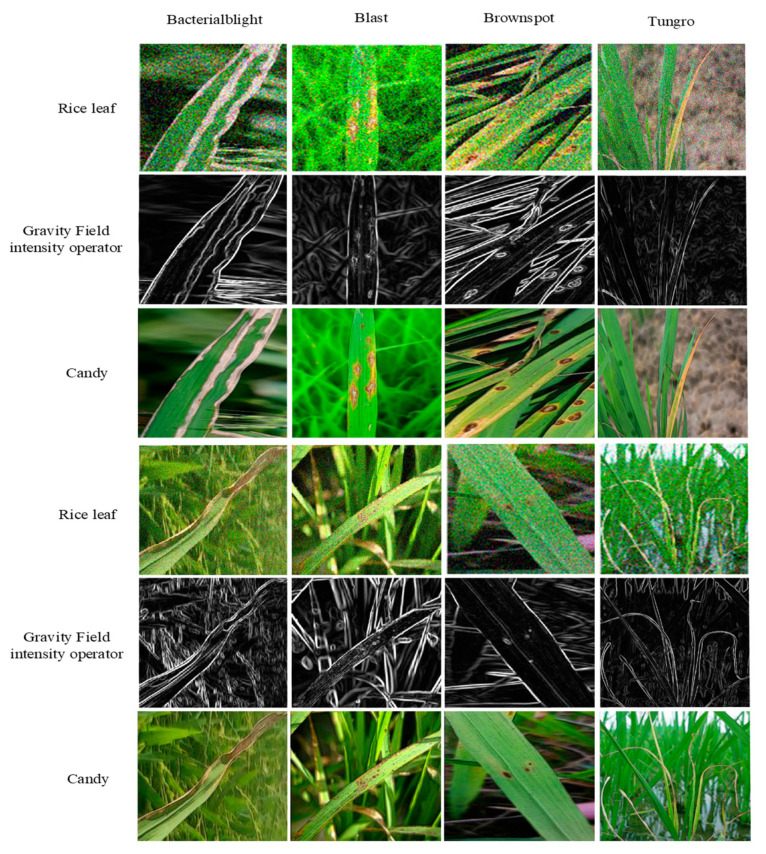
Picture utilizing the gravitational edge detection algorithm and Candy algorithm for processing rice leaf images.

**Figure 5 plants-12-02225-f005:**
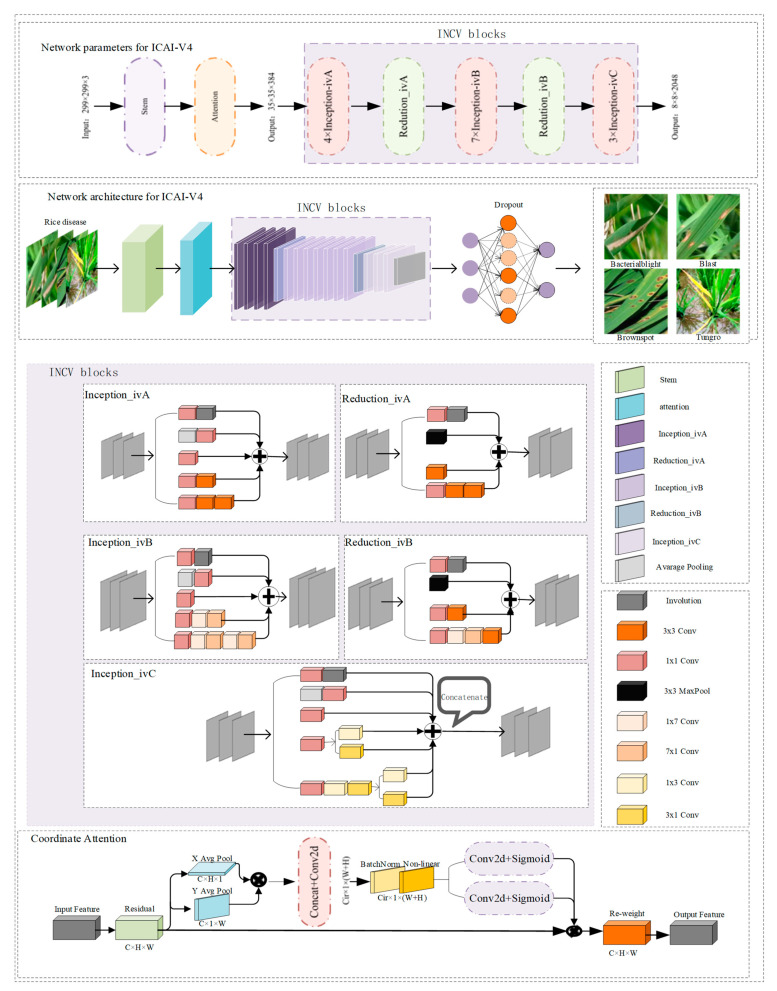
Block diagram of ICAI-V4 model. It includes INCV blocks and coordinate attention network structure.

**Figure 6 plants-12-02225-f006:**
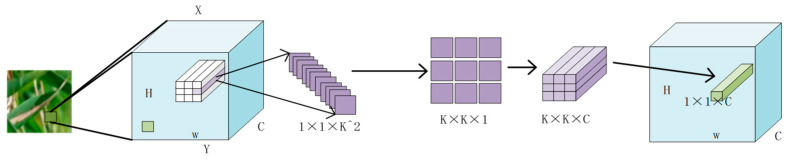
Schematic diagram of involution.

**Figure 7 plants-12-02225-f007:**
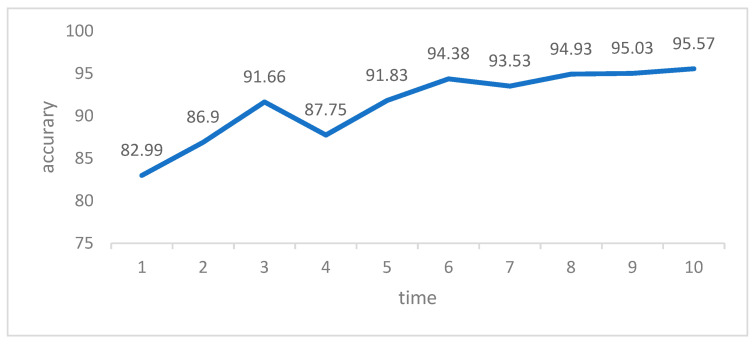
Accuracy of 10-fold cross-validation training results.

**Figure 8 plants-12-02225-f008:**
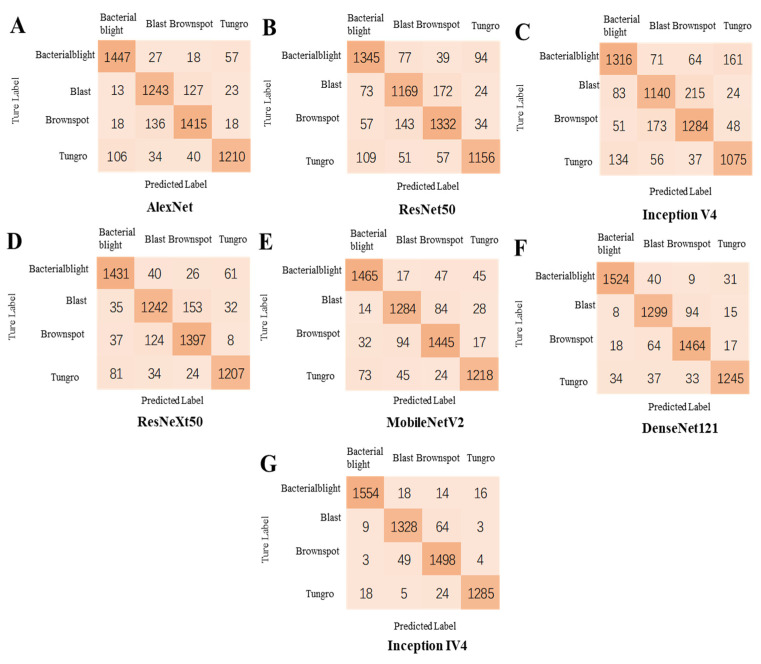
Classification confusion matrix of citrus diseases by different models (the models corresponding to the (**A**–**G**) confusion matrix are: AlexNet, ResNet50, Inceptionv4, ResNeXt, MobileNetv3, DenseNet121, and ICAI-V4).

**Table 1 plants-12-02225-t001:** Number and proportion of various rice disease images.

Disease Type	Kaggle	Mendeley Data	Expanded	Sum	Percentage (%)
Bacterial blight	1584	935	145	2551	26.70%
Blast	1440	894	289	2531	24.28%
Brown spot	1600	872	569	2845	26.97%
Tungro	1308	654	525	2314	22.05%
Total	5932	3355	1528	10,241	100%

**Table 2 plants-12-02225-t002:** Variable name and its meaning.

Symbol	Description
σ	The parameters of the Gaussian filter
$E_{x}\left[i,j\right]$	Approximation of the first partial derivative in the X direction
$E_{y}\left[i,j\right]$	Approximation of the first partial derivative in the Y direction
$G_{x}$	The image gradient operator in the x direction
$G_{y}$	The image gradient operator in the y direction
$\left|\left|M\left(i,j\right)\right|\right|$	Image gradient size
θ (x,y)	The azimuth of the image gradient
$\vec{E_{total}}$	Intensity of gravitational field
$\vec{E_{x}}$	The component of the gradient in the X direction
$\vec{E_{y}}$	The component of the gradient in the Y direction
E	Gradient magnitude
θ	Gradient azimuth
$E_{\mathrm{ave}}$	The average of the magnitude of the gradient
$T_{h}$	The threshold of a pixel
H	Involution kernel height
W	Involution kernel width
K	kernel size
$\varphi_{i,j}$	An index set in the neighborhood of coordinates $\left(i,j\right)$
$X_{\varphi_{i,j}}$	A patch in the feature map contains $X_{i,j}$
$y_{c}\left(i,j\right)$	Coordinate Attention Block output Y
TP	The number of samples that were predicted to be positive
FP	The number of samples that are actually negative, but predicted to be positive
FN	The number of samples that are actually positive, but predicted to be negative
TN	The number of samples that are predicted to be negative

**Table 3 plants-12-02225-t003:** Explanations for acronyms.

Acronym	Full Term
AI	Artificial Intelligence
CA	Coordinate Attention
FN	False Negative
FP	False Positive
GIS	Geographic Information System
IoT	Internet of Things
IT	Information Technology
TN	True Negative
TP	True Positive

**Table 4 plants-12-02225-t004:** Accuracy of rice disease classification after each change of INCV block structure.

Network Structure	Accuracy
Baseline	81.27
Inception_ivA	82.51
Reduction_ivA	82.93
Inception_ivB	85.14
Reduction_ivB	85.94
Inception_ivC	87.74

**Table 5 plants-12-02225-t005:** Classification model prediction results table.

Confusion Matrix		Prediction	
		True	False
Practical	True	TP	FN
	False	FP	TN

**Table 6 plants-12-02225-t006:** Hyperparameter settings.

Hyperparameters	Value
Learning rate	0.001
Epoch	20
Momentum	0.9
Batch size	32
Optimizer	RMSprop

**Table 7 plants-12-02225-t007:** Classification accuracy of different networks before and after image enhancement and dataset enhancement.

Image Enhancement Methods	Original Dataset (%)	Extended Dataset (%)
No image enhancement	81.27	89.26
Image enhancement	86.48	95.57

**Table 8 plants-12-02225-t008:** Ablation experimental results.

Model	InceptionV4	AlexNet	ResNeXt
Baseline	81.27	89.68	89.07
CA	89.06	91.25	91.21
INCV	87.74	92.14	90.13
Leaky ReLU	84.14	90.23	89.91
CA+INCV	94.25	94.35	91.25
CA+Leaky ReLU	91.24	93.13	90.63
INCV+Leaky ReLU	90.25	92.53	91.25
CA+INCV+Leaky ReLU	95.57	93.23	92.14

**Table 9 plants-12-02225-t009:** Performance of ICAI-V4 and other network models before and after adding noise to the dataset.

Methods	Accuracy (Test Set without Noise) (%)	Accuracy (Test Set Added Noise) (%)
AlexNet	89.68	84.72
ResNet50	84.48	80.67
Inception V4	81.27	79.31
ResNeXt50	89.07	87.45
MobileNetV2	91.31	89.14
DenseNet121	93.31	93.23
ICAI-V4	95.57	95.25

**Table 10 plants-12-02225-t010:** Classification performance evaluation of ICAI-V4 and other networks.

Methods	Evaluation Indicators	Rice Disease			
		Bacterial Blight	Blast	Brown Spot	Tungro
AlexNet	Precision (%)	93.41%	88.40%	89.16%	87.05%
	Recall (%)	91.35%	86.31%	88.43%	92.50%
	*F*1-score (%)	92.36%	87.34%	88.79%	89.69%
ResNet50	Precision (%)	86.49%	81.29%	85.05%	84.19%
	Recall (%)	84.91%	81.18%	83.25%	88.37%
	*F*1-score (%)	85.69%	81.23%	84.14%	86.22%
Inception V4	Precision (%)	81.63%	77.97%	82.51%	82.56%
	Recall (%)	83.08%	79.16%	80.25%	82.18%
	*F*1-score (%)	82.34%	78.56%	81.36%	82.36%
ResNeXt50	Precision (%)	91.84%	84.95%	89.20%	89.67%
	Recall (%)	90.34%	86.25%	87.31%	92.27%
	*F*1-score (%)	91.08%	85.59%	88.24%	90.95%
MobileNetV2	Precision (%)	93.07%	91.06%	90.99%	89.55%
	Recall (%)	92.48%	89.16%	90.31%	93.11%
	*F*1-score (%)	92.77%	90.09%	90.64%	91.29%
DenseNet121	Precision (%)	95.01%	91.73%	91.50%	95.18%
	Recall (%)	96.21%	90.20%	91.50%	95.18%
	*F*1-score (%)	95.06%	90.95%	91.50%	95.18%
ICAI-V4	Precision (%)	97.00%	94.58%	96.39%	96.47%
	Recall (%)	98.10%	92.22%	93.62%	98.24%
	*F*1-score (%)	97.54%	93.38%	94.98%	97.34%

**Table 11 plants-12-02225-t011:** Detailed information about the dataset.

Dataset	Category	Total	Available
PlantVillage	38	55,400	https://www.kaggle.com/datasets/hiyash99/plantvillage (accessed on 24 March 2023)
Stanford cars	196	16,185	https://www.kaggle.com/datasets/jutrera/stanford-car-dataset-by-classes-folder (accessed on 24 March 2023)
ImageNetDogs	120	20,580	http://vision.stanford.edu/aditya86/ImageNetDogs/ (accessed on 24 March 2023)

**Table 12 plants-12-02225-t012:** Public dataset test results.

Dataset	Accuracy (%)
PlantVillage	96.25
Stanford cars	94.67
ImageNetDogs	97.24

## Data Availability

All data are presented in the article.

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
