# Peer review of "An Accurate Classification of Rice Diseases Based on ICAI-V4"

_plants, 2023, doi:10.3390/plants12112225_

Round 1

Reviewer 1 Report

The article "An accurate classification of rice diseases based on ICAI-V4", addresses a subject of great agricultural interest, is well written with good presentation and an adequate structure.

The introduction has an excellent bibliographic review, on the subject, the methodologies and experimental design were well outlined and the results and their explanation are very clear.

The conclusions are supported by the results. As it contributes to an advance in knowledge on the subject under study, we are in favor of its publication in Plants.

Author Response

Thank you for your letter dated May 15. We were pleased to know that our work was rated as potentially acceptable for publication in Journal, subject to adequate revision. Thank you for your affirmation of this paper, I am very honored. Accordingly, we have uploaded a copy of the original manuscript with all the changes highlighted by using the track changes mode in MS Word.We would like also to thank you for allowing us to resubmit a revised copy of the manuscript.

Reviewer 2 Report

This paper proposes an image enhancement algorithm that utilizes a gravitational edge detection algorithm to improve edge feature highlighting and noise reduction in rice diseases in combination with a new neural network (ICAI-V4) that enhanced feature capture and overall model performance.

Generally, lacks a good contextualization of the problem and a real scope. It would be appropriate to add some paragraphs concerning for example precision agriculture (e.g., proximal sensing) and artificial intelligence.

I will consider the work for a publication only if a thorough review will be done.

Minor comments

Not all acronyms are spliced. Report either an initial table with all the explanations acronyms or always specify them before writing.

Generally, figure’s captions are not very detailed and explanatory.

To better understand the topic of the manuscript, I suggest changing the title to: “An accurate classification of rice diseases based on neural networks ICAI-V4”

Eliminate the “we” pronouns throughout the text.

Major comments

Introduction

This paragraph extensively describes the importance of rice monitoring during its growth, but no reference is made to the methodology used. In particular a section reporting the main application of the precision agriculture, such as for example, the proximal sensing, could be added. In addition, since you are analyzing the images with the neural networks ICAI-V4, a section on artificial intelligence should also be included.

Describes and reports application examples concerning proximal sensing, image analysis, etc.

There is no real aim. Insert it to make the activity more understandable.

Materials and methods

There is no paragraph on materials and methods in the text. Create one by moving the text of both the introduction and the results into it.

A flowchart that better explains the timing of sampling etc. would also be useful.

Conclusions

This section is very poor and not well organized and not very detailed.

Reviewer 3 Report

This paper needs major revision.

Round 2

Reviewer 2 Report

The manuscript could be accepted in the present form.

Author Response

Thank you for your letter dated May 28th. We are pleased to hear that our work has the potential to be accepted for publication in the Journal, provided that it undergoes adequate revision. We appreciate your affirmation of this paper, and I am personally honored by it.

Reviewer 3 Report

I consider this article acceptable overall. However, I would like to point out to the authors a few small, quickly manageable changes.
